# Regulatory and Sensing Iron–Sulfur Clusters: New Insights and Unanswered Questions

Anna M. SantaMaria and Tracey A. Rouault *

Molecular Medicine Branch, Eunice Kennedy Shriver National Institute of Child Health and Human Development, Bethesda, MD 20892, USA; anna.santamaria@nih.gov
* Correspondence: rouault@mail.nih.gov

**Abstract:** Iron is an essential nutrient and necessary for biological functions from DNA replication and repair to transcriptional regulation, mitochondrial respiration, electron transfer, oxygen transport, photosynthesis, enzymatic catalysis, and nitrogen fixation. However, due to iron's propensity to generate toxic radicals which can cause damage to DNA, proteins, and lipids, multiple processes regulate the uptake and distribution of iron in living systems. Understanding how intracellular iron metabolism is optimized and how iron is utilized to regulate other intracellular processes is important to our overall understanding of a multitude of biological processes. One of the tools that the cell utilizes to regulate a multitude of functions is the ligation of the iron–sulfur (Fe-S) cluster cofactor. Fe-S clusters comprised of iron and inorganic sulfur are ancient components of living matter on earth that are integral for physiological function in all domains of life. FeS clusters that function as biological sensors have been implicated in a diverse group of life from mammals to bacteria, fungi, plants, and archaea. Here, we will explore the ways in which cells and organisms utilize Fe-S clusters to sense changes in their intracellular environment and restore equilibrium.

**Keywords:** iron–sulfur clusters; iron metabolism; iron sensing; IRE; PAIR; Aft1/2; IRP1/2; RirA; Yap5; ACO1; IREB2; HSCB





## 1. Introduction

Fe-S clusters are comprised of iron and inorganic sulfur and are one of the most ancient components of living matter on Earth. They are integral cofactors for physiological function in all domains of life [1–4]. Fe-S clusters act as cofactors for reduction–oxidation (redox) reactions that can change the structure and, therefore, the function of proteins when they bind and/or change oxidation states [5,6]. In addition to their structural and catalytic roles, they may also function as sensors of iron and oxygen levels, interfacing between the intracellular environment and proteins involved in metabolic pathways [7–11]. Despite their ancient lineage and their key roles in many diverse biological functions, Fe-S clusters were not discovered and subsequently characterized until the late 1950s and early 1960s [2,3,12] when spectroscopic techniques and technologies were developed that enabled researchers to observe their intrinsic magnetic properties [1,13] (Figure 1). Since then, and especially in the last decade, numerous Fe-S-containing proteins have been discovered in mammalian systems; however, due to the inherently labile nature and oxygen sensitivity of these cofactors, there are potentially many more proteins dependent upon Fe-S clusters than previously imagined [13] (Figure 1).

Recently, DNA replication and repair enzymes have been found to contain Fe-S clusters [14–24]. For example, the helicase–nuclease DNA2, glycosylases, helicases, base excision repair enzymes, and nuclear replicative DNA polymerases were found to ligate Fe-S cluster cofactors. Defects in Fe-S cluster biogenesis cause a concurrent increase in mitochondrial and nuclear DNA damage and, ultimately, genomic instability [17,25]. In addition to being largely unstable cofactors as mentioned above, one of the main threats to

DNA integrity is the oxidation of nucleic acid bases. As iron often contributes to damaging oxidation through Fenton chemistry [26–29], it was initially surprising when many DNA replication and repair enzymes were found to require FeS cofactors to function, though their intrinsic properties could potentially increase DNA damage and genomic instability. In some instances, the Fe-S clusters act as redox switches that provide a means to detect oxidative stress and modulate replication [6,22,30,31]. Also, some Fe-S clusters provide electrons for DNA charge transfer, which is then used by DNA repair enzymes to signal one another and search for DNA lesions [6,22]. Most recently, several Fe-S clusters have been discovered to be integral cofactors involved in the viral replication of Severe Acute Respiratory Syndrome Coronavirus 2 (SARS-CoV-2), the pandemic-causing, causative agent of COVID-19 [32,33] (Figure 1). Three integral Fe-S cluster cofactors have been discovered in two SARS-CoV-2 proteins thus far. Two Fe-S clusters were found in the SARS-CoV-2 RNA-dependent RNA polymerase (RdRp-nsp12), which is responsible for genomic replication and transcription, and one Fe-S cluster was discovered in the SARS-CoV-2 RNA helicase (nsp13) [33]. These clusters represent interesting novel broad spectrum anti-viral targets that are susceptible to oxidative damage and potentially unlikely to become refractory to targeted degradation [34].

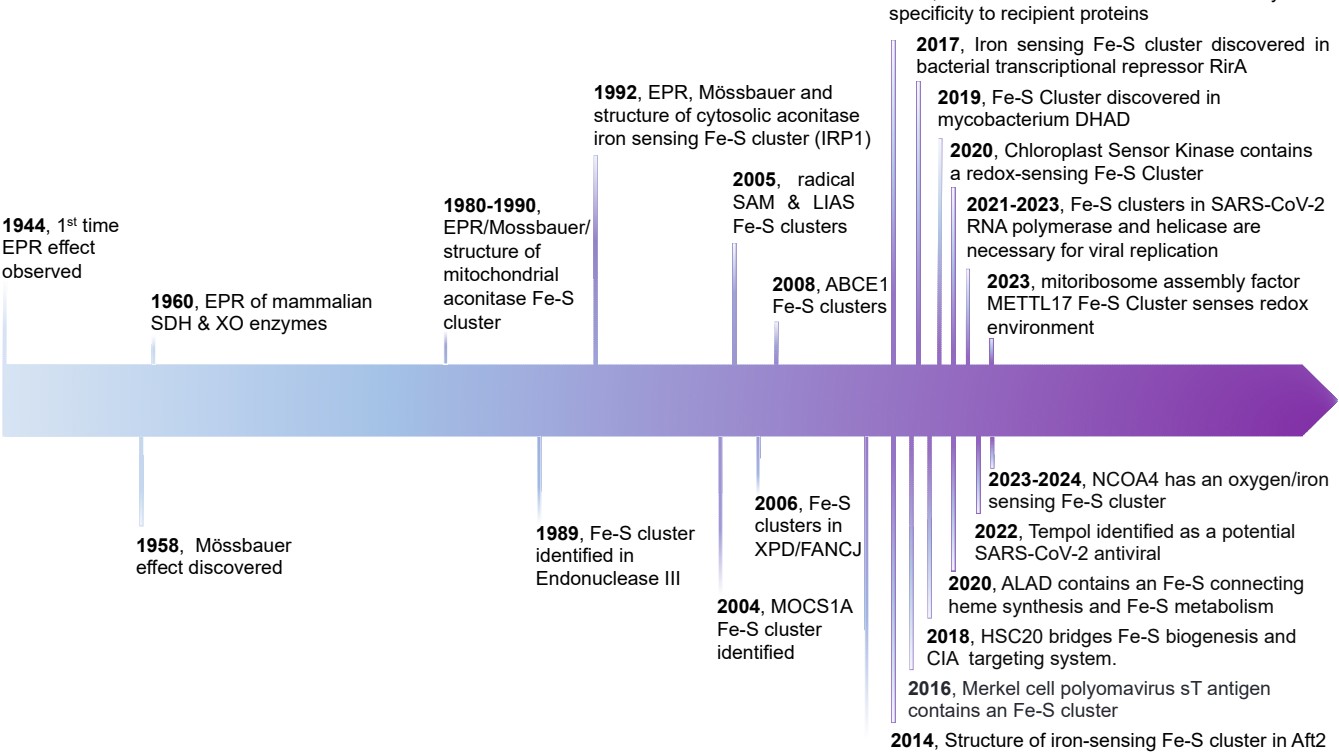

**Figure 1.** A timeline illustrating select discoveries in the Fe-S cluster field ranging from the discovery of EPR and Mössbauer effects to recent discoveries of environmental-sensing Fe-S clusters. Since their discovery in 1960, Fe-S clusters have been found to perform a variety of essential functions in all domains of life. With new scientific techniques and greater awareness of their important role within biology, the pace of discovery has increased exponentially.

Utilizing Fe-S clusters as cofactors in aerobic organisms is risky for the same reason it has been so difficult to discover new Fe-S cluster-containing proteins. Fe-S clusters readily decompose upon exposure to oxygen [5,13], which leads to one of the largest unresolved questions in the field: Why did evolution favor continued dependence on Fe-S clusters after the increased oxygen levels in the atmosphere changed to largely disfavor their stability? When life began on earth, there was plenty of available iron and sulfur for easy and spontaneous incorporation into enzymes in the form of Fe-S cluster cofactors [5]. However,

as oxygen levels rose, more iron was oxidized to less bioavailable forms while the lability and oxygen sensitivity of Fe-S clusters became a larger problem. This paradox suggests that the utilization of Fe-S cluster cofactors has been under positive selection pressure, and Fe-S cofactors perform an essential role that other, less labile cofactors cannot replicate. Some of the most important Fe-S cluster functions include sensing the intracellular environment and regulating cellular processes in response [10,13,35].

## 2. FeS Cluster Structure, Geometries, and Biogenesis

Iron–sulfur clusters exhibit a variety of structures and geometries depending on the number of iron and sulfur atoms in the cluster and the coordinating residues present in the protein. The coordinating amino acids found in most Fe-S cluster proteins are largely comprised of cysteine and histidine, but occasionally aspartate, glutamate, lysine, serine, and threonine are also coordinating residues [36,37]. The most common iron–sulfur cluster geometries found in biological systems are [2Fe-2S], [4Fe-4S], and [3Fe-4S] (Figure 2, **left**) [3,38,39]. Rhomboid [2Fe-2S] clusters consist of two iron atoms bridged by two amino acids residues; cubane [4Fe-4S] clusters consist of four iron atoms coordinated by four inorganic sulfur atoms and ligated by four amino acids forming a cubane-like structure with a central cavity, whereas pyramidal or triangular [3Fe-4S] clusters consist of three iron atoms coordinated by four inorganic sulfur atoms ligated by three amino acid ligands. Additionally, more complex clusters can be formed by combining the basic clusters together. For the most part, these complex FeS clusters are found in nitrogenases (Figure 2, **right**) [40–42].

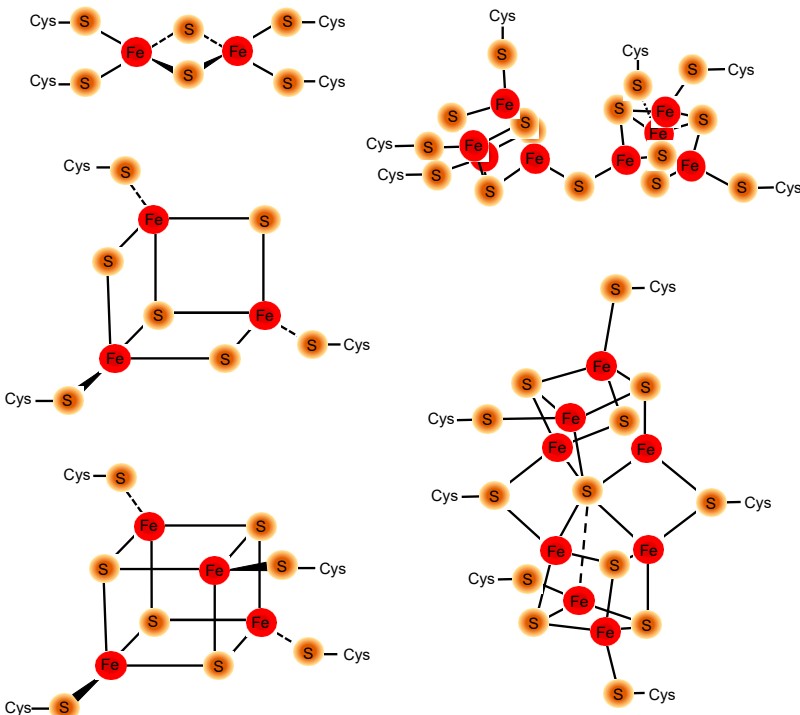

**Figure 2.** The most common FeS cluster geometries found in proteins are [2Fe-2S], [3Fe-4S], and [4Fe-4S] (**left**). Other more complex clusters can be found, especially in nitrogenases (**right**).

The transition metal iron is key to one of the main underlying functions of FeS clusters, the ability to accommodate reversible binding of a single electron and to enhance the movement of single, unpaired electrons through a relay system, such as that found in the respiratory complex I [43]. Fe-S clusters' unique capacity to accept and donate unpaired electrons in a non-energy-intensive, non-disruptive manner has cemented their place in key biological functions across the cell. The most notable examples of electron relay

systems are found in multiprotein complexes involved in mitochondrial respiration and photosynthesis [10,11,44,45].

In addition to their ability to aid in enzymatic catalysis via accepting or donating electrons, FeS clusters are also heavily implicated in cellular response to iron and oxygen. This is possibly due to the FeS cluster's inherent susceptibility to destabilization and degradation by a number of oxidation agents commonly found in the cell (superoxide, nitric oxide, etc.). The susceptibility of the cluster to degradation via oxidation is highly dependent on its location within the protein (solvent exposed or internal) [46]. The more solvent exposed the cluster is, the more opportunity there is for oxidative damage and degradation followed by the loss of the cluster. The other possibility is that FeS clusters are not synthesized and ligated in iron-sensing proteins in low-iron conditions.

The Fe-S cluster biogenesis machinery is highly conserved between eukaryotes and prokaryotes [11]. In mammalian cells, initial Fe-S cluster biogenesis occurs in parallel in both the mitochondrial matrix and cytosol [1,10,47–53]. The biogenesis of nascent Fe-S clusters is a highly energy-intensive process that requires multiple proteins. Fe-S cluster proteins are de novo assembled on scaffold protein ISCU [51,54–57]. Inorganic sulfur atoms are supplied by the cysteine desulfurase NFS1, which requires ISD11 to stabilize the interaction and the cofactor pyridoxal phosphate [58–60]. The release of sulfur atoms from cysteine and ligation to ISCU are promoted by the transient binding of frataxin. Sulfur combines with two $Fe^{2+}$ atoms and two reducing equivalents to form a [2Fe-2S] cluster on ISCU [61,62]. The iron necessary for cluster formation originates from an incompletely characterized pool of available iron, sometimes referred to as the "chelatable or labile iron pool" [63,64]. Ligands for available iron may include glutathione [65], cytosolic iron chaperones PCPB1 and BOLA2 [66–68], and perhaps ATP, citrate, GAPDH, and other proteins that are not yet characterized. In the mitochondria, ISCA1, ISCA2, and IBA57 proteins have been implicated in the conversion of [2Fe-2S] clusters to [4Fe-4S] [69]. There is evidence that NFU plays an important role in the maturation from [2Fe-2S] clusters to [4Fe-4S] clusters in both the mitochondrial matrix and the cytosol [55], but more details about the formation of cubane [4Fe-4S] clusters from the initial [2Fe-2S] building blocks formed on ISCU are needed. The newly formed Fe-S cluster can then be transferred to recipient proteins via a chaperone/cochaperone system. Briefly, holo-ISCU binds to the chaperone/cochaperone pair, HSC20 (aka HSCB) and HSPA9. HSPA9 is an ATPase that uses the energy from ATP to drive a conformational change in the Fe-S cluster transfer complex, enabling the complex to deliver Fe-S cofactors to specific recipients. An important advance in understanding how recipients are correctly identified emerged from studies of SDHB [43]. A three amino acid motif in recipient proteins that consists of an aliphatic amino acid in position one, followed by a large hydrophobic residue, either phenylalanine or tyrosine, and ending with a positively charged amino acid, either arginine or lysine, fits into a pocket in HSC20, which has been defined by mutagenesis [70]. Such motifs, known as LYR-like motifs, aid in the specificity of Fe-S cluster delivery to recipient proteins [70]. Specifically, recipient proteins are bound to the transfer complex when the nascent Fe-S cluster is released, leading to the safe, efficient, and specific transfer of the cluster from the biogenesis machinery to the recipient proteins [50]. The fact that the transfer process is guided by direct interactions between biogenesis machinery and Fe-S insertion into the recipients likely ensures that the vulnerable nascent Fe-S cluster will not be degraded by unprotected exposure to solvents and oxygen. HSC20 stimulates the ATPase activity of HSPA9, resulting in HSC20 dimerization via its J domain. Dimerized HSC20-ISCU may then bind to the cytosolic Fe-S platform protein, CIAO1, via CIOA1's LYR motif. CIAO1 is a part of a large multimeric iron–sulfur cluster delivery complex including MMS19, and FAM96B. Apo Fe-S cluster proteins likely interact with the large complex, likely due to the presence of a LYR-like motif, to receive their clusters. They then remain in the cytosol or are translocated to the nucleus to participate in DNA metabolism.

Much of our knowledge about FeS biogenesis arose from studies on bacteria, in which an operon devoted to FeS biogenesis was discovered [71,72]. Homology and processes

were sufficiently conserved so that bacterial studies informed many studies performed in eukaryotes, including yeast, plants, and mammalian cells and other organisms.

## 3. Human Iron Metabolism

Iron is efficiently recycled and, therefore, moves throughout the body into different tissues and cell types in a tightly controlled cycle. Typically, only 1 to 1.5 mg of dietary iron needs to be absorbed per day to replace iron loss from epithelial sloughing and minor blood loss in order to maintain sufficient iron stores to support normal physiological processes within the human [73]. Non-heme iron is absorbed from the diet by polarized gut enterocytes through the apical Divalent Metal Transporter 1 (DMT1, also known as NRAMP2, DCT1, or SLC11A2) [74], and secreted into the blood on the basolateral side of the epithelium by Ferroportin (FPN1, also known as IREG1, MTP1, or SLC40A1) [75] where ferrous iron is oxidized by membrane-bound hephaestin [76] and bound by transferrin (Tf) that circulates in the bloodstream [77]. Diferric transferrin then circulates iron throughout the body, delivering most transferrin-bound iron to the bone marrow, where developing erythroid cells endocytose the transferrin after it binds to the transferrin receptor (TFRC, aka TfR1) on the surface of the plasma membrane, strip it of iron and then pump the iron in the endosome through DMT1 into the cytosol and eventually to the mitochondria [78]. Iron enters the mitochondrial matrix through the inner-membrane iron transporter, Mitoferrin (MFRN1, also known as SLC25A37) [79,80], where heme synthesis and iron–sulfur cluster biogenesis occurs [1,52,81,82]. Once the hemoglobinization process has started, erythroid cells begin the differentiation process [83,84] into mature red blood cells, which circulate throughout the body delivering oxygen from the lungs to other tissues and organs. When red blood cells become damaged, they are recycled by specialized macrophages primarily in the spleen, and also in the liver in cells known as Kupffer cells [85]. Macrophages phagocytose the damaged red blood cells, strip the iron from heme and secrete it into the blood stream via the iron exporter, FPN1 [86–89], where transferrin once again binds iron and the cycle continues [90,91]. If perturbations due to environmental or genetic causes arise at any point in this process, the tightly controlled system can break down, resulting in symptomatic disease. For instance, when macrophages lack inducible heme oxygenase, they are unable to defend themselves against heme toxicity, and macrophage death causes a severe disease in mice and human patients, which can be overcome when normal functional macrophages are introduced [92–94].

While iron is an essential nutrient that is required for many important functions such as oxygen transport, electron transfer, oxidation–reduction reactions, and catalytic activity in metalloproteins, it can also be very toxic when it builds up in excess within the body and cells [95–98]. Iron is an efficient catalyst for biological reactions due to it redox capabilities, and it is, therefore, a cofactor in many enzymes. Unfortunately, the same properties that make it indispensable to life also lead to its toxicity. When too much free iron overwhelms a biological system, it can lead to the production of reactive oxygen species (ROS) and toxicity caused by the formation of superoxide followed by the formation of a hydroxyl radical [63,99–101]. Hydroxyl radicals lead to DNA and protein damage as well as lipid peroxidation, which can cause genetic mutations, necrosis, and tissue damage [95,102–104]. Therefore, even though iron deficiency is a worldwide problem, iron uptake, utilization, and storage mechanisms within biological systems have evolved to be highly regulated phenomena. Iron uptake is regulated by a complex system of iron-sensing proteins in concert with transcriptional and translational regulation because there is no efficient mechanism for the human body to release iron once it is acquired [95,105–107].

In addition to its potential to cause toxicity, iron is at the forefront of host–pathogen interactions [108–111]. Iron is a required nutrient for both pathogens and their hosts, leading to a tug-of-war between pathogens, the microbiome, and the host. This competition for iron causes each group to balance its efforts to obtain sufficient iron while simultaneously preventing bacterial competitors from successfully acquiring iron [108–112]. Often, microbes biosynthesize siderophores to scavenge for iron in their environment [110,113–115],

while hosts sequester iron away from pathogens by binding it with calprotectin in blood cells such as neutrophils [116,117], storing it in the protein ferritin [118], or allowing it to circulate throughout the body, tightly bound to the protein transferrin [112,119,120]. It is very rare for any iron to be unbound to some type of chelator within the body. This sequestration of iron away from potentially pathogenic microorganisms is known as nutritional immunity [109,110,112,121]. Excess transferrin helps to keep pathogens at bay by minimizing the presence of non-transferrin-bound iron (NTBI) [108–110,120,122–124].

Because of iron's important role in biological systems, and the need for tight regulation of iron within the body, it makes sense that hundreds of diseases are associated with impaired iron metabolism. Disorders of iron metabolism are the most common genetic deficiencies worldwide, and because iron is such an integral metal ion for many biological processes, iron overload or deficiency can affect almost every system and process of the body [77,119,125–127]. Currently, there are more than 25 known hereditary diseases caused by deficiencies in iron regulation and homeostasis, and additional disorders can be caused by acquired loss of protein function due to chronic illness, inflammation, and other environmental factors [127–129].

## 4. Intracellular Iron and Oxygen Sensing

There is significant cross-talk between oxygen-sensing and iron-sensing pathways [8,9,35,130]. Proteins involved in both pathways can influence each other's activity. The availability of iron can influence the assembly and stability of Fe-S clusters in oxygen-sensing proteins. Additionally, when oxygen levels are high, Fe-S clusters in iron response and iron regulatory proteins can be degraded, resulting in changes in the Fe-S cofactor that normally result during a low-iron state [35,130–138]. Cells, therefore, use Fe-S clusters as highly sensitive and efficient cofactors to sense and respond to intracellular iron and oxygen levels. Mammalian, bacterial, and yeast cells utilize Fe-S cluster-containing iron regulatory proteins for intracellular iron sensing. In the broadest sense, iron regulatory proteins, such as IRP1 (aka ACO1) in humans [1,7,10,107,139–142], IscR/RirA in bacteria [8,27,143–147], and Aft1 in yeast [148–151] sense iron levels via the presence or absence of Fe-S clusters. In iron-replete conditions, the ligation of Fe-S cluster cofactors is favored. However, in cases of iron depletion or oxidative stress, Fe-S cluster incorporation is disfavored due to decreased Fe-S biogenesis, or oxidative degradation, respectively. The ligation of iron–sulfur clusters causes functional and conformational changes of the protein's activity and structure, allowing for the modulation of expression of downstream iron metabolic proteins.

### 4.1. Mammalian Iron-Sensing Iron–Sulfur Clusters

In mammalian cells, iron transport and storage proteins are transcriptionally, translationally, and post-translationally regulated to maintain physiological intracellular labile (or chelatable) iron levels while avoiding functional iron overload and toxicity [106]. Specifically, levels of the iron transporter Divalent Metal Transporter 1 (DMT1), the iron sequestration and storage protein ferritin (heavy, FTH1 and light, FTL1 chains), the sole iron efflux protein ferroportin (FPN1), and the iron uptake mediator transferrin receptor 1 (TfR1) are translationally or post-transcriptionally regulated through short hairpin iron response elements (IREs) located at the 5′- and 3′-untranslated regions (UTR) of the corresponding mRNA transcripts that impair translation (5′UTR IRE) or prolong mRNA half-life (3′UTR IRE) [10]. Iron-sensing iron response proteins (IRP1 and IRP2) bind to these IREs to block translation (5′-IRE: FTH1, FTL1, FPN1, HIF2α, ACO2, and ALAS2) or stabilize mRNA (3′-IRE: DMT1, TfR1, CDC14A, and MRCKα) under iron starvation [7,137,138,152–163].

Upon iron stimulation and binding, the IRPs dissociate from the mRNAs, reversing their described effects. Transcriptional regulation is achieved through the transcriptional activator, hypoxia-inducible factor 2-alpha (Hif2α, aka EPAS1), which is degraded after oxygen- and iron-mediated proline hydroxylation [106]. Interestingly, Hif2α activates the transcription of a transcript of Fpn1 that evades IRE-mediated translational repression under iron deprivation conditions [136,154].

The mechanisms that IRP1/2 utilize to sense and react to intracellular iron levels are elegant (Figure 3). IRP1 exists in two mutually exclusive forms, either as a cytosolic aconitase or as an RNA-binding protein [7,163,164]. The dual nature of IRP1 allows this unique protein to sense iron levels within the cytosol by conditionally ligating an Fe-S cluster in its active site cleft. In iron-replete conditions, IRP1 acquires a [4Fe-4S] cluster. The holo-IRP1 enzyme acts as a cytosolic aconitase, isomerizing citrate and isocitrate [10,139]. In iron starvation conditions, the Fe-S cluster of IRP1 is absent, and IRP1 becomes an IRE-binding protein [35,83,107,142]. The activation of IRE-binding activity increases cytosolic iron levels by reducing iron sequestration in ferritin and by increasing transferrin receptor activity [7]. These two major responses are accomplished by repressing the translation of ferritin and stabilizing mRNA levels of TFRC (TFR1), which allows the increased synthesis of the transferrin receptor. Whereas IRP1 utilizes an Fe-S cluster to switch its activity from aconitase to IRE binding, IRP2 (aka IREB2) does not contain an Fe-S cluster and does not exhibit aconitase activity [1,77,141,165,166]. This divergence in activity and cluster ligation is surprising due to the high sequence similarity (~60%) that IRP1 and IRP2 share [7,167,168]. IRP2, therefore, only exhibits iron regulatory properties via its IRE-binding activity. Despite not containing an Fe-S cluster directly, IRP2 levels are still influenced by an oxygen-sensing Fe-S cluster [165,169]. It is not surprising that two proteins would have the same regulatory function but differ in how they are regulated, as many genes in the mammalian genome likely arose during the whole genome duplication of an evolutionary precursor [170] enabling organisms to develop systems that are robustly supported by multiple pathways [171], some of which have mechanistically diverged over time. For instance, the degradation of IRP2 when iron levels are high is a more robust response than turning off IRE-binding activity through the insertion of an Fe-S cluster cofactor that occludes the IRE-binding site.

The protein FBXL5 recruits ubiquitin ligase to specifically bind to and target IRP2 for ubiquitination and proteosomal degradation [140,165,169,172–174]. FBXL5 does this in an iron- and oxygen-dependent manner. When iron levels are low, the destabilization and degradation of FBXL5 renders it unable to bind to and target IRP2 for proteasomal degradation [165]. The [2Fe-2S] cluster in the C-terminal domain of FBXL5 is highly redox-sensitive [169]. FBXL5 can, therefore, utilize this redox-sensitive [2Fe-2S] cluster to sense oxygen levels and regulate IRP2 protein levels [169]. When the [2Fe-2S] cluster is in the 2+ oxidized state, a conformational change occurs that allows for a rearrangement of the c-terminus and binding of FBXL5 to IRP2, allowing for the ubiquitination and proteasomal degradation pathway to move forward. Therefore, while IRP2 itself does not directly ligate an intracellular environment-sensing Fe-S cluster, it is still regulated by one that occupies a different position in the overall regulatory scheme. Thus, the activity of each IRP depends on the status of an Fe-S cofactor which determines how much IRE-binding activity each IRP contributes to the regulation of its nine major target transcripts, either through regulating the rate of translation, or by stabilizing the transcript to allow increased protein synthesis. Interestingly, parallel IRE/IRP systems have been discovered in other eukaryotes including protozoans like *Trichomonas vaginalis* [175].

Recently, NCOA4, the master regulator of ferritinophagy, has gotten a lot of attention from the iron biology field. Ferritinophagy is the process via which ferritin is degraded in the autolysosome so that the iron it has stored can be recycled for use within the cell. Aberrant ferritinophagy has been linked to neurodegeneration and pancreatic cancer [28,176–179]. NCOA4 was known to bind iron and regulate ferritinophagy in response to iron levels, but until recently it was unknown how NCOA4 sensed iron [28,177]. NCOA4 has now been proposed as both an oxygen sensor [180] and an intracellular iron sensor [181] due to its ligation of an Fe-S cluster. These new discoveries highlight the inextricable link between Fe-S clusters and the regulation of intracellular regulation of iron metabolism.

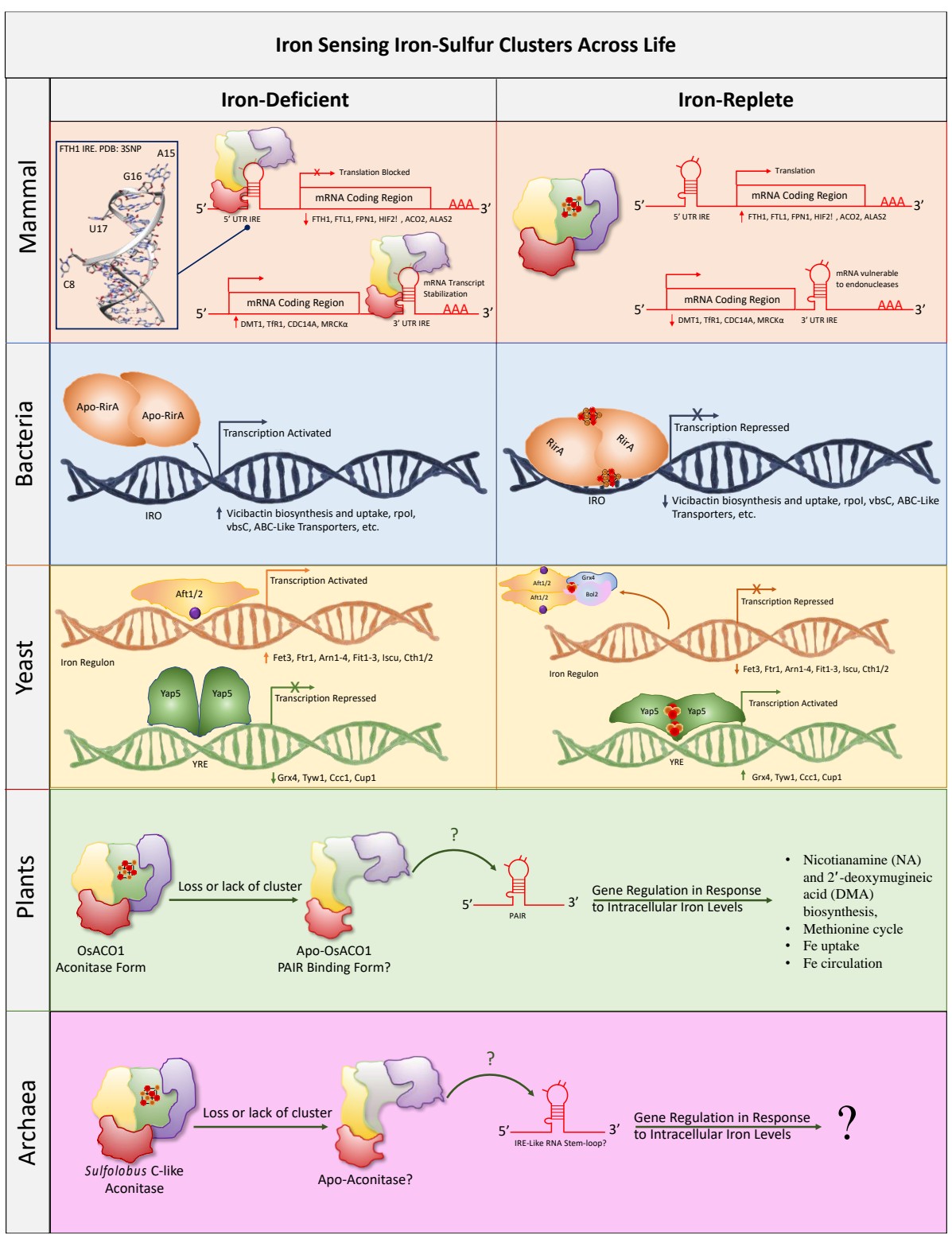

**Figure 3. Different iron-sensing Fe-S cluster strategies across many forms of life.** The labile nature of Fe-S clusters is taken advantage of by many organisms to sense and react to intracellular iron levels. Mammalian cells utilize IRP1 to react to intracellular iron levels. When iron levels are low, IRP1 exists in its apo RNA-binding form. When iron levels are replete, IRP1 exists primarily in its holo aconitase form. Bacteria utilize a number of FeS cluster sensors. Recently, RirA was discovered to bind the IRO under iron-replete conditions in order to block the transcription of iron acquisition genes. Similar to the mammalian and bacterial systems, yeast utilize iron-sensing FeS clusters to increase or decrease the expression of genes involved in iron homeostasis. In low-iron conditions, Aft1/2 does not oligomerize

with Grx4 and Bol2 and, therefore, does not ligate its FeS clusters. This allows Aft1/2, with a Zn atom ligated (purple), to bind to the yeast iron regulon and upregulate genes required for iron uptake. In iron-replete conditions, Aft1/2 ligate FeS clusters via interaction with Grx4 and Bol2. The binding of the clusters causes Aft1/2 to disengage from the yeast iron regulon and expression is downregulated. Conversely, Yap5 blocks gene expression in its apo form and upregulates translation in its holo form. In iron-replete conditions, holo-Yap5 binds to the Yap-Responsive Elements (YRE) and increases the transcription of genes involved in iron sequestration and storage. When YAP5 is in its apo form, it still binds the YRE but it blocks transcription. Recently, evidence that plants utilize a dual function aconitase has come to light. Similar to IRP1 rice aconitase, OsACO1 has been implicated in binding to RNA stem-loop structures called Plant ACO-Interacting RNA (PAIR). Archaea may also utilize a dual functional aconitase to sense and react to intracellular iron levels; however, more studies are needed to confirm this hypothesis.

### 4.2. Bacterial Iron-Sensing Iron–Sulfur Clusters

Iron is a limiting reagent for microorganism growth, both pathogenic and free-living. As is the case with mammalian and multicellular organisms, iron is both necessary for survival but also highly toxic at high concentrations [182]. Bacteria, therefore, require the ability to sense intracellular and extracellular iron to maintain optimal iron concentrations for growth. Bacteria have a large arsenal of iron acquisition techniques which range from simple membrane-bound iron transporters to small molecule siderophore scavengers [113,115,182–184]. Iron uptake needs to be tightly controlled to prevent iron overload and subsequent toxicity.

Bacteria use many environmental-sensing Fe-S clusters in order to react to intracellular and extracellular conditions. These sensing Fe-S clusters have been covered extensively in the literature [9]. One recently discovered example of an iron-sensing Fe-S cluster protein in bacteria is the iron-responsive *Rhizobium* protein rhizobial iron regulator (RirA) [27,185]. Mutations in RirA cause deregulation, and, therefore, constitutively high transcription rates of many operons involved in iron acquisition. *Rhizobium* is a genus of nitrogen-fixing, gram-negative bacteria that associate with the root systems of plants in structures called root nodules [186]. The plant, often in the legume or flowering plant family, and the rhizobium bacteria have a symbiotic relationship wherein the rhizobium utilizes its nitrogenase to convert atmospheric nitrogen in the soil to bioavailable nitrogen for the plant to use to optimize growth conditions [186]. The plant supplies the bacterial colonies with nutrients through their association with the root system, which allows both organisms to thrive. Similar to IRP1, RirA exists both in an apo and holo [4Fe-4S] cluster form [143]. Like the mammalian system, there is redundancy to modulate such an important physiological function, and rhizobium has an additional non-Fe-S cluster-binding co-regulator similar to IRP2, known as the iron response regulator (Irr), which aids in the modulation of intracellular iron levels [183]. When RirA is in its holo [4Fe-4S] cluster form, it represses iron uptake by binding to the iron response operator (IRO) box, a DNA sequence present in promoters of multiple genes [8,27,37,143]. Binding to the IRO box by RirA suppresses the transcription of iron uptake genes (Figure 3). When RirA loses its cluster due to low-iron and/or aerobic conditions, it undergoes a conformational change and can no longer bind to the IRO box, which then allows unfettered transcription and the expression of numerous iron acquisition genes [8,27,143–145].

Numerous other bacterial regulatory proteins utilize Fe-S cofactors to sense levels of nitric oxide, oxidative stress, Fe-S cluster sufficiency, and oxygen levels to regulate the transcription of genes needed to combat environmental challenges. Responses range from the upregulation of DNA repair enzymes and siderophore synthesis, to dramatic global changes such as sporulation, increased motility, and biofilm production [187]. Since Fe-S cofactors are often lost or replaced by zinc during aerobic purification, it is reasonable to predict that more bacterial Fe-S cluster sensors will be discovered in the future [13,188].

### 4.3. Yeast Iron-Sensing Iron–sulfur Clusters

Like mammalian and bacterial intracellular iron-sensing systems, yeast utilize iron-sensing Fe-S clusters to regulate iron metabolism. Yeast react to iron deficiency quickly and efficiently via transcriptional regulation implemented by three main transcription factors, Aft1, Aft2, and Yap5. Aft1/2 are paralogs that share 26% sequence identity and regulate yeast iron metabolism via turning on and off the yeast iron regulon [148,150,151]. Specifically, similarly to IRP1 which regulates translation, when iron levels are low, apo Aft1/2 monomers bind to DNA sites to increase the transcription of members of the iron regulon [148,189,190]. When cytosolic iron levels rise, Aft1/2 ligate a [2Fe-2S] cluster, causing a conformational change that does not allow for DNA binding, thus shutting off the transcription of the iron regulon and thereby reducing iron uptake. Aft1/2 gain their cluster by forming a complex with Grx4/5 and Bol2 [150] (Figure 3). Members of the regulon aid in the acquisition of iron via cellular membrane iron transporters (e.g., FET3/FTR1), siderophore binding/uptake (e.g., ARN1-4, FIT1-3), Fe-S biogenesis machinery (ISCU), and mRNA-binding regulatory proteins (CTH1/2) [189,191]. The iron-transporting complex Fet3Ftr1 is required for growth on low-iron media and is one of the main ways in which yeast acquire iron [127,192,193]. ARN1-4 are responsible for transporting siderophore–iron chelates into the cell and these transporters are, therefore, important for iron scavenging in scarce iron conditions [127,194,195]. CTH1/2 target and degrade RNA transcripts of nonessential proteins that require copious amounts of iron for their function. A more extensive list of genes regulated by Aft1/2 has been compiled (Kaplan & Kaplan 2009) [189]. Opposing Aft1/2, Yap5 constitutively binds to the Yap-Responsive Element within the genome [148,190,191,196–199]. When iron levels are low, Yap5 represses the transcription of iron storage and sequestration proteins. When intracellular iron levels are high, YAP5 ligates a [2Fe-2S] cluster, which changes its conformation to promote the transcription of iron storage and sequestration proteins among other proteins necessary to adapt to high intracellular iron [197,198]. Specifically, Yap5 regulates a yeast glutaredoxin (Grx4), a wybutosine-modified tRNA synthesis enzyme (Tyw1), a vacuolar iron–copper transporter Ccc1, and a copper chelator (metallothionein Cup1) [150]. The proteins upregulated by Yap5 aid in protecting the cell from protein and DNA damage due to iron overload by sequestering iron in vacuoles (Ccc1), turning off the iron acquisition pathway via the delivery of iron–sulfur clusters to Aft1/2 (Grx4), binding copper to halt ferrireductases from aiding iron import (Cup1), and increasing the fidelity of tRNAs to ensure the proper translation of proteins when iron concentrations are high (Tyw1). Though yeast model systems are among the most studied model organisms for a very good reason, more questions remain about how they regulate iron metabolism and interact with their environment.

### 4.4. Plant Iron-Sensing Iron–Sulfur Clusters

Although iron is the most abundant element, by mass, in the earth, the vast majority of iron is in a biologically inaccessible, insoluble ferric hydroxide state [5,200]. Plants, like the other organisms above, deal with nutritional iron deficiency in areas where bioavailable iron is reduced. Conversely, plants need to be able to reduce iron uptake in iron-rich soils. Unlike more motile forms of life, most plants cannot physically move to find the perfect balance of nutrients once they have established themselves. Lack of motility means that they need to have the ability to effectively sense and react to changes in their environment at the molecular level. Therefore, plants need to be able to sense the iron availability or lack thereof in their environment as well as intracellularly.

Currently, there are hints that Fe-S clusters may be involved in iron-sensing pathways, but no definitive evidence has been found to date [201]. As mentioned above, IRP1 has a dual function as an RNA-binding, iron response protein, and alternatively, as a cytosolic aconitase. Interestingly, a recent study in rice by Senoura et al. seems to promise the possibility of an iron-sensing Fe-S cluster in rice [202]. When one of three aconitase genes, *OsACO1*, is knocked down in rice, changes in iron homeostasis and iron-responsive genes are observed. OsACO1 is an aconitase that binds a [4Fe-4S] cluster and is expressed

ubiquitously throughout the plant. Rice heterozygous for OsACO1, *OsACO1*$^{(+/-)}$, have reduced aconitase activity as well as a reduced expression of genes involved in iron trafficking and regulation. The homozygous deletion of OsACO1 is embryonically lethal. A reduction in the nicotianamine (NA) synthase gene, *OsNAS2*, lowers plant NA synthesis, which reduces the long-distance transport of iron due to decreased NA [202,203]. NA is an iron chelator that plants utilize to enhance iron trafficking throughout the plant [202], analogous to the function of transferrin in iron trafficking and circulation throughout the body in humans. Additionally, the lowered gene expression of OsYSL15, a protein involved in iron uptake in roots [204], and OsIRO2, a transcription factor involved in iron homeostasis [205], is observed in the roots of *OsACO1*$^{(+/-)}$ rice plants. There is also evidence that OsACO1 has RNA-binding activity. Similar to mammalian IREs to which IRP1/2 bind, the RNA probes bound by OsACO1 in a gel-shift assay contain an RNA stem-loop structure composed of a seven-nucleotide loop on a stem of five-paired nucleotides with an unpaired bulge on the 5′ strand of the stem. In IRP1/2-IRE binding, the loop, five base-paired stem, and unpaired bulge have been shown to be vital for IRP1/2 recognition of the IRE [202] (Figure 3). The authors called this new RNA stem-loop structure Plant ACO-Interacting RNA element or PAIR. Subsequent studies confirming that the [4Fe-4S] in rice OsACO1 is utilized as an iron-sensing Fe-S cluster are likely to be interesting and will help fill in the missing link of how plant cells physically sense iron to better adapt to their environment.

*4.5. Archaea Iron-Sensing Iron–Sulfur Clusters*

Archaea are an ancient form of prokaryotic life that is estimated to have diverged over 3.9 billion years ago [206–208]. The archaeal iron–sulfur biogenesis pathway is not highly conserved between archaea species or between archaea, bacteria, and eukaryotes [209]. While archaea have more in common with eukaryotes than prokaryotes, the species that do have proteins associated with canonical Fe-S cluster biogenesis machinery most likely acquired them via lateral gene transfer from bacteria throughout their evolutionary journey [210,211]. Like the majority of life, most species of archaea rely heavily on Fe-S clusters in a variety of processes including electron transfer performed in large part by ferredoxins [212]. A subtype of archaea called Methanogens is thought to contain the highest percentage of Fe-S cluster-containing proteins per genome of any other organism [213]. Currently, it is thought that the anaerobic environments that methanogens inhabit contribute to their ability to contain such a uniquely large percentage of Fe-S cluster proteins in their proteome [213]. It is, therefore, surprising that iron–sulfur clusters have not been more heavily implicated in intracellular iron sensing in archaea, especially considering their heavy reliance on Fe-S cluster cofactors. Though a paper published in 2002 suggested that the aconitase of *Sulfolobus solfataricus* could be acting as an iron-sensing Fe-S cluster similarly to IRP1 [214] (Figure 3), no subsequent confirmational studies have been published. The domain archaea comprises a vast and diverse group of organisms that inhabit some of the most extreme environments on earth. Archaea, therefore, represent a treasure trove in which new discoveries and yet-to-be elucidated biological breakthroughs may be uncovered.

## 5. Conclusions, Future Perspectives, and Remaining Questions

Tight control of iron metabolism and, therefore, intracellular iron sensing are critical to all forms of life. Iron-sensing Fe-S clusters represent a rapid and highly sensitive cofactor for the detection of iron levels. When these ancient cofactors are incorporated (or not) into transcription factors and post-transcriptional modulators, they allow cells to sense and react to their intracellular environment through the alteration of protein expression. While iron–sulfur clusters are ubiquitous and essential cofactors in all domains of life, many unanswered questions remain. Fe-S cofactors are hard to study and isolate due to the same properties that enable them to be good iron and oxygen sensors. Their labile nature and propensity to degrade in the presence of oxygen often cause them to disappear when

proteins are purified on the bench in the presence of oxygen. Therefore, it is very possible that Fe-S cluster-containing proteins are even more ubiquitous than they are currently thought to be. They are hiding in plain sight, waiting for an anaerobic chamber and a determined researcher to reveal their existence. Recently, essential Fe-S clusters were found to be necessary for the causative agent of the pandemic-causing COVID-19 (SARS-CoV-2) virus to replicate. It is very possible that even more viruses depend on Fe-S clusters to replicate their genetic material, or perhaps sense and modulate the metabolism of the cells they are infecting. How many more human-disease-relevant Fe-S cluster cofactors are yet undiscovered?

Interestingly, there is very little data on potential iron-sensing Fe-S clusters in plants and archaea. With archaea being some of the most ancient forms of life on earth and known to contain many Fe-S cluster proteins, it would not be surprising if at least some species utilize Fe-S clusters to sense iron and oxygen levels within their cytosol. An additional large and multipart remaining question regarding sensing iron–sulfur clusters is the following: How do the ligating proteins switch between their apo and holo forms? In order to sense the intracellular environment, do the proteins lose their clusters, or are the Fe-S cofactors never incorporated due to lack of iron? If they lose their cluster due to low iron levels, are they able to regain a cluster under iron-replete conditions? How does the stability of the cluster compare to its turnover due to damage and degradation? When does normal acquisition of the cluster occur? While the protein is folding or after the protein is folded?

The field of iron–sulfur cluster biology has come a long way over the past few decades with an explosion of newly discovered Fe-S cluster-containing proteins. These discoveries open new chemical and biological vistas to study these fascinating, ancient, and essential metal cofactors. As always, questions remain and new discoveries await!

**Author Contributions:** Conceptualization, A.M.S. and T.A.R.; methodology, A.M.S. and T.A.R.; writing—original draft preparation, A.M.S. and T.A.R.; writing—review and editing, A.M.S. and T.A.R. All authors have read and agreed to the published version of the manuscript.

**Funding:** This work was supported by the Intramural Research Program of the National Institutes of Health (NIH), Eunice Kennedy Shriver National Institute of Child Health and Human Development (NICHD).

**Data Availability Statement:** No new data were created or analyzed in this study. Data sharing is not applicable to this article.

**Conflicts of Interest:** The authors declare no conflict of interest.

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
