# Peer review of "Regulatory and Sensing Iron–Sulfur Clusters: New Insights and Unanswered Questions"

_inorganics, doi:10.3390/inorganics12040101_

Round 1

Reviewer 1 Report

Comments and Suggestions for Authors

In this review the authors describe how iron-sulfur clusters are utilized to sense cellular environments/stimuli, to regulate iron metabolism, and to maintain the correct equilibrium between cellular iron requirements and iron toxicity. The iron-sulfur cluster sensing processes occurring from humans to yeast, plants, bacteria and archeae are briefly discussed. The iron-sulfur proteins involved in their sensing mechanisms are reported and how they act is discussed. Overall, the review reads well and clearly describes where still research studies are required to clarify the role of iron-sulfur clusters in the sensing/regulatory processes. My only concern is related to that the lack of a more detailed description of the coordination chemistry and reactivity of the iron-sulfur sensing clusters. This part is indeed not presented at all by the authors in the reported examples, but I believe that it would be very important to help to identify when and how iron-sulfur clusters are able to have an iron-sensing activity. For example, are there common structural/molecular features associated to iron-sulfur clusters in sensing or all iron-sulfur clusters bound to proteins can be effective in sensing ?

Comments on the Quality of English Language

The quality of English is very good and just requires minor editing.

Author Response

We thank the reviewer for their insightful suggestions and taking the time to review our manuscript. We have added a small paragraph about the location of iron sensing iron sulfur clusters. We specifically point out that solvent exposed ligation is common to better sense the intracellular oxidative and iron environment. Additionally, we state that sensing iron sulfur clusters can sense by not being ligated in the first place during times of iron starvation.

Reviewer 2 Report

Comments and Suggestions for Authors

This paper by Rouault and coauthor summarized the generation, regulation and function of iron-sulfur clusters, especially as iron sensors in mammals, bacteria, yeast, and plants. In general, the paper is informative and helpful to the field. The following issues need to be addressed before it is accepted.

1.       The abstract could be more informative about the content of this paper. The current version only mentioned this in the last sentence.

2.       Including the timeline as Figure 1, the paper has only two figures, which is very rare as a review paper. Description only with words for several portions of the paper makes it difficult to accept and less friendly for nonexpert readers. These include the structures of Fe-S and the biogenesis process.

3.       The legend of Figure 2 is too simple and needs to be revised.

4.       At the beginning of the paper the authors mentioned that Fe-S was used as sensors for iron and oxygen, but the paper only introduced iron sensors. Could also introduce cases as oxygen sensors?

5.       Some sentences in the paper look very arbitrary and need to be more careful, such as the first sentence in the 4.4 section, which is wrong in common sense.

Author Response

We thank you for taking the time to evaluate our manuscript and for providing your insightful suggestions. We have made a number of changes to address each of your concerns. Specifically, we altered the abstract to name the organisms we would be covering in the review, we added a figure with biological relevant geometries of FeS clusters, expanded the figure 2 caption, and made the first sentence in section 4.4 more clear by rewording it adding an additional reference. We discussed a few oxygen sensing FeS clusters throughout the manuscript including FBXL5 and NCOA4. There is also a short discussion on how and why iron and oxygen metabolisms are connected and correlated. 

Round 2

Reviewer 1 Report

Comments and Suggestions for Authors

The authors have nicely modified the text discussing the functional implications of iron sulfur sensing clusters with respect to their location on the protein structure.  

Reviewer 2 Report

Comments and Suggestions for Authors

The authors have addressed my concerns. Publish as is.